# Reaction with ROO• and HOO• Radicals of Honokiol-Related Neolignan Antioxidants

**DOI:** 10.3390/molecules28020735

**Published:** 2023-01-11

**Authors:** Nunzio Cardullo, Filippo Monti, Vera Muccilli, Riccardo Amorati, Andrea Baschieri

**Affiliations:** 1Dipartimento di Scienze Chimiche, Università di Catania, V.le A. Doria 6, 95125 Catania, Italy; 2Istituto per la Sintesi Organica e la Fotoreattività (ISOF), Consiglio Nazionale delle Ricerche (CNR), Via Gobetti 101, 40129 Bologna, Italy; 3Dipartimento di Chimica “G. Ciamician”, Università di Bologna, Via S. Giacomo 11, 40126 Bologna, Italy

**Keywords:** antioxidant activity, honokiol, neolignans, peroxyl radicals, hydroperoxyl radicals, quinones’ regeneration, radical reactions, reaction mechanisms, hydrogen atom transfer

## Abstract

Honokiol is a natural bisphenol neolignan present in the bark of *Magnolia officinalis*, whose extracts have been employed in oriental medicine to treat several disorders, showing a variety of biological properties, including antitumor activity, potentially related to radical scavenging. Six bisphenol neolignans with structural motifs related to the natural bioactive honokiol were synthesized. Their chain-breaking antioxidant activity was evaluated in the presence of peroxyl (ROO•) and hydroperoxyl (HOO•) radicals by both experimental and computational methods. Depending on the number and position of the hydroxyl and alkyl groups present on the molecules, these derivatives are more or less effective than the reference natural compound. The rate constant of the reaction with ROO• radicals for compound **7** is two orders of magnitude greater than that of honokiol. Moreover, for compounds displaying quinonic oxidized forms, we demonstrate that the addition of 1,4 cyclohexadiene, able to generate HOO• radicals, restores their antioxidant activity, because of the reducing capability of the HOO• radicals. The antioxidant activity of the oxidized compounds in combination with 1,4-cyclohexadiene is, in some cases, greater than that found for the starting compounds towards the peroxyl radicals. This synergy can be applied to maximize the performances of these new bisphenol neolignans.

## 1. Introduction

Lignans and neolignans are two groups of dimeric compounds widely distributed into the plant kingdom and biosynthesized through the shikimic acid pathway (Appendix A). The feature of these molecules is a peculiar dimeric structure originated by a ß, ß,’-linkage between two phenyl propane units, C6C3, characterized by different degrees of oxidation in the side-chain and distinctive substituents occurring in the aromatic rings [1]. Their biosynthesis is originated from oxidative coupling involving phenyl propanoid units by enzymes such as laccase, peroxidase, or a cytochrome P450 [2], thus furnishing a wide range of dimeric compounds with different structures (Appendix A) [3]. For nomenclature purposes, the C6C3 units are treated as propylbenzene. When the linkage occurs between positions C-8 and C-8′ of two C6C3 units, the compound is a “lignan”; in a “neolignan”, the dimer is formed through a linkage involving two C6C3 units in positions different from C-8 and C-8′ (C-8-C5′, C-5-C-5′, etc.) [4]. In turn, given the high number of possible combinations, lignans are classified into eight groups according to structural patterns (Figure 1), whereas neolignans are classified into fifteen subgroups, indicated as NL1 to NL15 (Figure 2) [4].

Lignans and neolignans are polyphenols often studied for their antioxidant behavior [5]. Some non-exhaustive examples are in the following (Figure 3). Pinoresinol is one of the most representative lignans, found in sesame seeds and extra-virgin olive oil; it is considered a high-value-added product with antioxidant activity, useful in chemoprevention [6]. (*-*)-Arctigenin is one of the main components of *Arctium lappa* whose extracts have been employed in Japanese Kampo medicine for antioxidant properties with benefits to human health [7]. A group of glucosidic dihydrobenzofuran neolignans have been also studied as antioxidants [8].

Magnolol and honokiol are neolignans with a diphenyl core (bisphenol neolignans, Figure 3); their structure is peculiar, and some authors consider the phenolic rings linked through a C–C bond a privileged structure which allows interaction with a variety of biological targets [9,10]. These two bisphenolic neolignans are present in the bark of *Magnolia officinalis* and other *M.* spp, whose extracts have been employed in oriental medicine to treat several disorders such as gastrointestinal disorders, anxiety, stress and allergic and cardiovascular diseases [11]. In addition to the antioxidant property, the two bisphenols have shown a number of biological properties, such as neuroprotective [12], antiviral [13], anti-inflammatory [14] and antitumor activities [15,16,17].

For this reason, recent works have been dedicated to the synthesis of new analogues inspired by magnolol and honokiol with the purpose of enhancing their biological activities [16,18,19,20].

In this frame, the antioxidant behavior of magnolol and its isomer honokiol (**1**) has been deeply studied [21,22,23]. Interestingly, the two compounds have shown different antioxidant profiles derived from the position of the OH groups. In organic solvents, magnolol was more active as a peroxyl (ROO•) radical quencher than honokiol, because of the stabilization of the phenoxyl radical by an intramolecular H-bond [23]. Indeed, the study of the kinetics of ROO• and HOO• radical trapping is of great relevance because these radicals are responsible for the propagation of the oxidative chain during lipid peroxidation [24], and are implicated in important biochemical processes such as ferroptosis [25].

The synthesis of bioinspired natural antioxidants represents a strategy to gain new molecules showing stronger antioxidant capacity than natural leads. A similar or even better antioxidant profile has been observed for bioinspired derivatives of magnolol as measured by the rate constant of the reaction with ROO• radicals [26]. As a continuation of these investigations, six bisphenol neolignans with structural features resembling the natural bioactive honokiol (**1**) were synthesized and evaluated for their antioxidant behavior (Figure 4). In particular, we have designed honokiol-related compounds **2** and **3** to compare their antioxidant profiles with that of **1** and to understand a possible role in the oxidative processes arising from the presence of the ortho-allyl chain. Furthermore, supposing **2** and **3** will result in promising antioxidants, the presence of the 2-hydroxyethyl chain in *para* position to OH could allow the insertion of other functional groups to gain compounds with different physicochemical properties for future studies. Moreover, in a previous work, the bisphenol **8** (3,3′,5,5′-tetramethyl-[1,1′-biphenyl]-4,4′-diol) has shown a large rate constant for reaction with peroxyl radicals, arising from the conjugation of the radical on both aromatic rings, and from the presence of the methyl groups in the ortho position that reduce the bond dissociation enthalpy of OH. In addition, this compound showed a stoichiometric coefficient of 1.9 in the autoxidation of styrene, indicating that it transfers the second O−H atom to a second peroxyl radical. Based on these findings, we have designed bisphenols **5** and **6** and thus catechol **7** and its methylated analogue **4**. The kinetics of peroxyl and hydroperoxyl radical trapping was studied using the inhibited autoxidation method which provides, with respect to other simplified methods based on the decay of colored radicals, a more solid prediction of efficacy in real conditions [27].

## 2. Results and Discussion

### 2.1. Synthesis of Bisphenol Neolignans **2**–**7**

Bisphenol neolignans **2**–**6** were synthesized following a previously described strategy [28] based on a Suzuki-Miyaura cross-coupling step between a suitable aryl halide and 4-hydroxy-phenyl boronic acid, to build the biphenyl skeleton of compounds **2**, **4** and **5**. Subsequently, S_N_2 reaction in presence of allyl bromide followed by Claisen rearrangement in mild conditions allowed us to isolate the *C*-allyl derivatives **3** and **6**. The spectroscopic data of these compounds are in agreement with those previously reported [28] (Figure 1).

On the contrary, the synthesis of catechol **7** is reported herein for the first time. As depicted in Figure 2, the bisphenol **4**, obtained by Suzuki coupling, was converted into the catechol analogue employing hypervalent-iodine chemistry, according to literature reports on similar substrates [16,26,29]. In particular, 2-iodoxybenzoic acid (IBX) was prepared following the protocol of Frigerio M. et al. [30]. IBX was employed in slight excess with respect to **4** (1.2 equiv) and in THF. These represent the optimal conditions to gain the oxidative demethylation of a guaiacol group, thus achieving the bis-*ortho*-quinone intermediate which is converted into the final catechol when a saturated Na_2_S_2_O_4_ solution is added to the mixture. The compound was isolated after column chromatography with a 38% yield. Details and spectroscopic data (NMR and MS data) of the new bisphenol **7** are reported in the Materials and Methods section. Notably, IBX is more environmentally friendly, if compared with other oxidizing agents based on heavy or toxic metals and/or requiring strong reaction conditions.

### 2.2. Kinetics and Stoichiometry of the Reaction with Peroxyl Radicals

The antioxidant activity of honokiol (**1**) and six inspired bisphenol neolignans **2**–**7** (AH) was evaluated by measuring the rate constant (*k*_inh_) for the reaction with peroxyl radicals (ROO•) that are responsible for oxidative chain propagation in many natural materials [31].
Initiator → R^•^(1)
R^•^ + O_2_ → ROO•(2)
ROO• + RH → ROOH + R^•^(3)
ROO• + ROO• → Non-radical products(4)
ROO• + AH → ROOH + A^•^(5)
ROO• + A^•^ → Non-radical products(6)
where, *R*_i_ is the rate of initiation (reaction 1). Equations (1)–(4) represent the autoxidation of substrate RH in the absence of antioxidants, while Equations (5) and (6) represent chain-breaking inhibition by antioxidant AH.

The inhibition rate constants (*k*_inh_) of antioxidants **1**–**7** (i.e., the rate constant of reaction 5), were determined by studying the inhibition of the thermally initiated autoxidation of cumene or styrene (RH) under controlled conditions using chlorobenzene or acetonitrile as the solvent [32].

All reactions were performed at 303 K using 2,2′-azobis(isobutyronitrile) (AIBN) as initiator and were followed by monitoring the oxygen consumption in an oxygen uptake apparatus based on a differential pressure transducer [27,32].

For the very first step, the rate of radical initiation produced by AIBN (*R*_i_) was determined in matched preliminary experiments using the inhibitor method, according to Equation (7)
*R*_i_ = *n* [AH]/τ(7)
where τ is the length of the inhibition time. Tocopherol’s mimic 2,2,5,7,8-pentamethyl-6-chromanol (PMHC) was used as reference antioxidant, with *n* = 2.

In the presence of effective antioxidants, substrate oxidation and oxygen consumption are much slower and a clear inhibition period is observed, as shown in Figure 5.

The rate constant for the reaction between ROO• radicals and **1**–**7** could be obtained from the rate of O_2_ consumption (the slope of the oxygen consumption) during the inhibited period from the known constants *k*_p_ and 2*k*_t_ for cumene (and styrene) chain propagation and termination, respectively, using Equation (8) where *R*_ox0_ and *R*_ox_ represent the O_2_ consumption rate in the absence and in the presence of the antioxidant, respectively (see Experimental Section and ref [32,33,34,35,36] for more explanations).
(8)$$\frac{R_{\mathrm{ox}0}}{R_{\mathrm{ox}}}-\frac{R_{\mathrm{ox}}}{R_{\mathrm{ox}0}}=\frac{nk_{\mathrm{inh}}{\left[AH\right]}_{0}}{\sqrt{2k_{t}R_{i}}}$$

The stoichiometric coefficient *n*, which represents the number of ROO• radicals trapped by each antioxidant molecule, is instead determined from the duration of the antioxidant effect. The values of *k*_inh_ and *n*, determined in chlorobenzene and acetonitrile, are reported in Table 1.

To fully solubilize the samples, 0.2% (*v/v*) methanol was added to all chlorobenzene and in acetonitrile solutions. For this reason, it was also necessary to re-evaluate, even if already reported in the literature [23], the rate constants of honokiol **1,** since it was used as a reference compound for all the other investigated molecules. Despite being present in a very small amount, methanol strongly affects the inhibition constant of honokiol **1** in chlorobenzene by decreasing it about three times (see Table 1 vs. ref [23]). Conversely, in acetonitrile, this effect is limited, since such solvent already forms hydrogen bonds with the OH groups of the antioxidant molecules. Additionally, in this series of experiments, the number of radicals trapped by honokiol **1** in chlorobenzene is equal to 2, while it nearly doubles in acetonitrile.

In PhCl, as previously demonstrated, the phenoxyl radical from honokiol **1** reacts with a second ROO• radical by formal H-atom transfer from an OH group, leading to the formation of the corresponding dienone **1ox** (Figure 3). In MeCN, on the other hand, the second OH group in the phenoxyl radical from honokiol **1** is H-bonded to the solvent, and therefore it is less available to be transferred to a second ROO• radical, as shown in Figure 3. As a consequence, the phenoxyl radical decays preferably through the addition of a second ROO• radical to the aromatic ring. The intact second phenolic ring is then available to trap two additional peroxyl radicals, similarly to monophenolic compounds but with a lower *k*_inh_ than the first OH, presumably because of the unfavourable electronic effect of the oxidized ring.

Antioxidants **2** and **3** are comparable to **1**. The inhibition constant is about half that of honokiol both in a non-polar solvent, such as PhCl, and in a polar solvent, such as MeCN. In compound **2,** the absence of alkyl groups lowers the stability of the phenoxyl radical intermediate, while electronegative O-atom in the hydroxyethyl chain has a small negative impact on the inhibition constant. In compound **3**, the allyl substituent present in an *ortho* position with respect to the -OH groups limits the H-atoms abstraction because of a OH---π interaction, as previously reported [23].

The number of radicals trapped by these last two compounds is the same as for the reference compound **1**. In MeCN, for the second phenolic ring, the stoichiometric coefficient could not be measured because the *k*_inh_ value is too low. For this reason, we only found *n* = 2.

Compound **4** has a different structure than honokiol. Due to the different position of a phenolic group in one of the two rings, it is not possible to obtain the corresponding quinonic form and therefore the two phenolic rings can be considered independent. Each ring traps 2 radicals with the addition of the peroxyl radical (Figure 4). The inhibition constant of the more reactive -OH group is completely comparable to that of compound **1**, while the second constant is about 10 times lower. For this reason, only the first can be observed and measured in MeCN.

Bisphenols **5** and **6** display relatively large *k*_inh_ values compared to honokiol, with the *k*_inh_ of **5** being about twice that of **6**. As already demonstrated for compound **3**, the presence of allyl substituents reduces the antioxidant activity of these compounds, due to hydrogen bonding between allylic and hydroxyl groups; such an effect is less noticeable in a polar solvent such as MeCN [23]. Since both **5** and **6** have a stoichiometric coefficient of about 2 in the inhibited autoxidation of cumene, it is expected that the phenoxyl radicals from both **5** and **6** transfer the second O−H atom to another peroxyl radical, as shown in Figure 5.

The inhibition shown by compound **7** is the highest of all the investigated compounds, as expected from the presence of a second OH group in *ortho*-position (catechol moiety) (Figure 6). In this case, the experiments were performed in styrene. Styrene is typically employed for studying strong antioxidants (i.e., with *k*_inh_ > 1×10^5^ M^−1^s^−1^), whereas cumene, thanks to its low *k*_p_ and 2*k*_t_ values, is suitable for weaker inhibitors [32]. The *k*_inh_ of **7** is two orders of magnitude greater than compounds **1**–**6**, but it corresponds to the trapping of only two ROO• radicals.

This behaviour is reminiscent of that observed in *ortho*-bisphenol derivatives [37,38]. It can be explained by considering that, after the trapping of the first two ROO• radicals, one of the two phenolic rings is converted into the corresponding *ortho*-benzoquinone, which has an unfavourable effect on the second phenolic ring, reducing its reactivity, possibly because of electron-withdrawing effect. In our case, the ortho-quinonic form **7ox_A_** is also in equilibrium with the dienonic form **7ox_B_**.

As shown in Table 1, the *k*_inh_ values decrease for all phenols when the polarity of the solvent is increased (i.e., in acetonitrile); this is known as the kinetic solvent effect (*KSE*), which occurs in case of H-atom abstraction from polar X−H bonds [39]. The decrease is more evident for **7**, as the *KSE* is 48, while it ranges from 1 to 3 for compounds **1**–**6**. Notably, this phenomenon has already been studied for other compounds having a catechol ring [40,41]. The solvent engages in H-bonds with phenolic OH groups, and decreases their reactivity towards the peroxyl radicals by preventing the formation of the H-atom transfer pre-reaction complex. As for the other compounds, the lowest values (about 1.5) are observed in molecules equipped with allyl groups. Indeed, such groups, already form H-bonds with hydroxyl groups, so the effect in MeCN is less evident. Accordingly, slightly higher values (about 2 or 3), are found for compounds having unsubstituted phenolic rings.

### 2.3. Hydrogen Atom Transfer from HOO• to Quinones

As described above, with the exception of compound **4**, the honokiol-inspired bisphenol neolignans, upon reaction with the peroxyl radicals in chlorobenzene, oxidize to the corresponding quinone forms **1ox**, **5ox** and **7ox** (see Figure 3, Figure 4, Figure 5 and Figure 6).

Autoxidation of 1,4 cyclohexadiene (CHD) to benzene is a well-known chain process, in which HOO• acts as a propagating radical (Equation (10)) [42,43].
CHD + XOO^•^ → CHD_−H•_ + XOOH     (X = H or R)(9)
CHD_−H•_ + O_2_ → benzene + HOO•(10)
XOO^•^ + HOO• → XOOH + O_2_     (X = H or R)(11)
Q + HOO• → QH^•^ + O_2_(12)
QH^•^ + HOO• → QH_2_ + O_2_(13)
QH_2_ + HOO• → QH^•^ + H_2_O_2_(14)
2 QH^•^ → Q + QH_2_(15)

Equations (12)–(15) explain the key reactions of the antioxidant activity of quinones, (Q = **1ox**, **5ox** and **7ox**), in the presence of 1,4-cyclohexadiene.

The addition of CHD in the peroxidation of oxidizable substrates (RH) partially changes the propagation chain-carrier from ROO• to HOO• (hydroperoxyl radical) since CHD itself is rapidly attacked by ROO• and releases HOO•. Such hydroperoxyl radicals can both propagate the oxidation reaction or be quenched by another HOO• or by a ROO• radical (self-termination or cross-termination).

To achieve a better understanding of the regeneration mechanism of phenolic antioxidants by CHD, experiments were conducted by injecting CHD 26 mM into the styrene autoxidation system after the phenolic antioxidant was consumed at the time that the substrate starts to oxidize again and approximately the whole compound is in the form of an oxidized product.

As shown in Figure 6D, the injection of CHD into the reaction, when sample **4** has been completely oxidized, provides only a very small increase in the inhibition (Figure 6D and Table 2).

On the other hand, in the presence of fully exhausted honokiol **1** or compounds **5** or **7**, the addition of CHD triggers a new inhibition period (Figure 6A–C and Table 2). We explain this effect by considering that the quinones formed as the final oxidized products (Q = **1ox**, **5ox** and **7ox**) can be reduced back to the starting antioxidant, confirming the mechanism suggested in Figure 3, Figure 4, Figure 5 and Figure 6. When catechols behave as antioxidants, quinones are easily formed upon oxidation and they are generally expected to be their final oxidized products. Although such a mechanism is less obvious for other polyphenolic compounds, we were able to demonstrate the formation of the corresponding quinones also for honokiol and the bisphenol **5**. Comparing the kinetic traces in Figure 6, we should notice that the rate of O_2_ uptake during the inhibition period is smaller for **7** than for **1** and **5**, demonstrating the superior antioxidant activity of the *ortho*-isomer (Table 2).

The reduction of all quinones is attributed to the release of HOO• during the autoxidation of CHD which acts as the reducing agent. This uncommon reducing behavior might be counterintuitive for a reputedly oxidizing radical, but it is supported by previous solid evidence [44,45,46].

From the slopes shown in Figure 6 (red lines vs. black lines), it is clear that the antioxidant effect of **1**, **5** and **7** is visibly lower than that of oxidized products **1ox**, **5ox** and **7ox** with CHD, except for compound **7** and its product, which are both high. Additionally, the inhibition length is clearly higher when HOO• radicals produced by the co-oxidation of CHD with styrene are present in the reaction environment.

Therefore, quinones formed by oxidation of **1**, **5** and **7** are effectively regenerated by HOO• radicals; this synergic antioxidant chemistry, exploiting CHD in combination with polyphenolic compounds, is more effective than traditional antioxidant systems.

### 2.4. Theoretical Calculation of Bond Dissociation Enthaplies

To rationalize the kinetic results, the preferred conformations and the dissociation enthalpies of the O–H bonds (BDE(OH)) were computed by DFT methods at the B3LYP-D3/6-31+G(d,p) level [47,48,49,50], using the SMD [51] implicit solvation model, as implemented in the Gaussian 09 [52]. The BDE(OH) values in chlorobenzene were obtained by using an isodesmic approach that consists of calculating the BDE difference between the investigated compounds and phenol (ΔBDE(OH)), and by adding this value to the known experimental BDE(OH) of phenol in benzene (86.7 kcal/mol) (Equation (16)).
BDE(OH) = 86.7 + ΔBDE(OH)(16)

The results of BDE(OH) calculations for both the hydroxyl groups present in compounds **1**–**7** are shown in Table 3. From these calculations, it is possible to recognize which phenolic ring is intrinsically more reactive towards peroxyl radicals, allowing us to know the structure of the corresponding semiquinone that is generated. While BDE(OH) alone is not a complete descriptor of *k*_inh_, nevertheless, for phenols having similar steric crowding around the reactive OH, a Evans-Polanji-type relationship between Log(*k*_inh_) and BDE(OH) can be observed.

In the case of compounds **1**–**7**, the fairly linear relationship (Figure 7) indicates that theoretical calculations account with reasonable accuracy the reactivity of ROO• radicals in non-polar media and confirms the previous structure–activity relationship discussion for the compounds studied in this manuscript.

The bond dissociation enthalpy of the O−H bond of semiquinones **1ox**, **5ox** and **7ox** was also calculated by DFT methods to confirm their reactivity towards hydroperoxyl radicals as observed in previous experiments, in the presence of CHD. The reaction of semiquinones with HOO• depends on the BDE of the phenolic O−H bond in the semiquinone. If the BDE is high, the quinone will react more easily with hydroperoxyl radicals and there will be an overall antioxidant effect on the system.

It is known that the semiquinone obtained from the reaction of 2,5-di-*tert*-butylhydroquinone with peroxyl radicals is able to react with molecular oxygen, dissolved in air-equilibrated solutions [53]. For this reason, the BDE(OH) of 1,4-semiquinone was calculated and used as a reference (65.3 kcal/mol). The semiquinones of compounds **1ox**, **5ox** and **7ox** have BDE(OH) values of 79.2, 75.2 and 73.6 kcal/mol, respectively. Since these values are greater than the BDE of 1,4-semiquinone, the quinones **1ox**, **5ox** and **7ox** are expected to have a fast reaction with HOO• radicals, while having a slow reverse reaction of the corresponding semiquinone with oxygen. The BDE(OH) order would predict that HOO• trapping decreases in the order **1ox** < **5ox** < **7ox <<** 1,4-benzoquinone. However, these BDE values, obtained by DFT methods, are only a theoretical prediction of the synergistic effect between CHD and the oxidized products; this does not take into account kinetic aspects, such as the stability in solution of these quinones, and the formation of a non-regenerable products obtained by adding peroxyl radicals to the phenolic rings, which have the effect of reducing the concentration of quinone available for the reaction with HOO•. Nevertheless, the comparison between the data obtained with DFT methods and those obtained in the above experiments fully rationalize the obtained results and confirm the proposed reaction mechanisms.

## 3. Materials and Methods

### 3.1. Materials

All chemicals were of reagent grade and were used without further purification. Where necessary, starting materials, namely aryl halides [28] IBX were freshly prepared as previously described [30]. Solvents were of the highest grade commercially available and were used as received. Commercially available honokiol **1** was purchased from TCI Europe (Milan, Italy), PMHC (2,2,5,7,8-Pentamethyl-6-chromanol) and AIBN were purchased from Sigma-Aldrich (Milan, Italy). Cumene, styrene and 1,4 cyclohexadiene were purified by double percolation through silica and activated alumina columns before use. AIBN was recrystallized from methanol and stored at −18 °C.

NMR spectra were acquired on a Varian Unity Inova spectrometer (Italy, Milan) operating at 499.86 (^1^H) and 125.70 MHz (^13^C). 1D and 2D NMR experiments (gHSQC, and gHMBC) were performed at 300 K. A high-resolution MS spectrum of **7** was run on a Q Exactive Orbitrap mass spectrometer (Thermo Fisher Scientific, Bremen, Germany) equipped with an ESI ion source operating in negative ion mode. Compound **7** was directly infused in the spectrometer, and a survey scan was performed from *m/z* 150 to 1000 at 140 k resolution.

### 3.2. Synthesis of Bisphenol Neolignans

Compounds **2**–**6**, used in the present investigation, were synthesized as described previously and as depicted in Figure 1 [28].

Compound **7** was synthesized for the first time as reported in the following. The purity of these compounds was verified by ^1^H NMR analysis.

The bisphenol **4** (62.6 mg, 0.24 mmol) was solubilized in THF (4 mL) and IBX (80.2 mg, 1.2 equiv.) was added under stirring to the solution. The mixture was stirred at rt for 3 h. Then, a saturated Na_2_S_2_O_4_ solution (4 mL) was added and the mixture was stirred at rt for 10 min. The crude of reaction was concentrated under vacuum to remove THF and the residue was diluted with EtOAc (20 mL) and partitioned with saturated NaHCO_3_ solution (3 × 10 mL). The organic layer was washed with brine solution and dried over Na_2_SO_4_. After filtration, the solvent was evaporated under vacuum. The pure product **7** was obtained after column chromatography on Diol silica gel (100 → 80:20 *n-h*exane/acetone) with 38% yield (22.3 mg). ^1^H-NMR (500 MHz, CDCl_3_): 7.12 (d, *J* = 8.2 Hz, 2 H, H-2B/H-6B), 6.83 (d, *J* = 8.2 Hz, 2 H, H-3B/H-5B), 6.77 (s, 1 H, H-5A), 6.69 (s, 1 H, H-2A), 5.15 (bs, 2 H, 3A-OH/4A-OH), 4.86 (bs, 1 H, 4B-OH), 2.41 (m, 2 H, C*H_2_*-7A), 1.44 (m, 2 H, C*H_2_*-8A), 0.79 (t, *J* = 7.3 Hz, 3 H, C*H_3_*-9A) ppm. ^13^C-NMR (125 MHz, CDCl_3_): 154.2 (C, C-4B), 142.5 (C, C-4A), 140.8 (C, C-3A), 134.2 (C, C-1B), 134.1 (C, C-1A), 133.3 (C, C-6A), 130.6 (CH, C-2B/C-6B), 117.1 (CH, C-2A), 116.0 (CH, C-5A), 114.8 (CH, C-3B/C-5B), 34.5 (CH_2_, C-7A), 24.6 (CH_2_, C-8A), 13.1 (CH_3_, C-9A) ppm. HRESIMS *m*/*z* 243.1049 [M-H]^-^ (calcd for C_15_H_12_O_3_, 243.2857).

### 3.3. Inhibited Autoxidation Studies

Autoxidation experiments were performed in a two-channel oxygen uptake apparatus, based on a Validyne DP 15 differential pressure transducer built in our laboratory [32]. The chain-breaking antioxidant activity of the title compounds was evaluated by studying the inhibition of the thermally initiated autoxidation of cumene (3.6 M) or styrene (4.3 M) in chlorobenzene and acetonitrile. In a typical experiment, an air-saturated mixture of the oxidizable substrate and the solvent, 1:1 (*v*/*v*) containing AIBN (0.05 M) as initiator was equilibrated with an identical reference solution containing an excess of PMHC so as to block any radical chain in the reference and avoid significant consumption of the antioxidant therein during the experiment. After equilibration, and when a constant O_2_ consumption was reached, a concentrated solution of the antioxidant (final concentration = 5–20 μM) was injected in the sample flask. The oxygen consumption in the sample was measured after calibration of the apparatus from the differential pressure recorded with time between the two channels. Initiation rates, *R*_i_, were determined for each condition in preliminary experiments by the inhibitor method using PMHC as a reference antioxidant: *R*_i_ = 2[PMHC]/τ, where τ is the length of the induction period. PhCl/Cumene *R*_i_ = 5.9 × 10^−9^ Ms^−1^; MeCN/Cumene *R*_i_ = 8.2 × 10^−9^ Ms^−1^; PhCl /Styrene *R*_i_ = 6.2 × 10^−9^ Ms^−1^. From the slope of oxygen consumption in the absence of antioxidant (−*d*[O_2_]/*d*t)_0_ = *R*_ox0_) and during the inhibited period (−*d*[O_2_]/*d*t) = *R*_ox_), *k*_inh_ values were obtained by using Equation (8). The 2*k*_t_ values of styrene and cumene at 303 K are 4.2 × 10^7^ and 4.6 × 10^4^ M^−1^s^−1^, respectively [54,55].

### 3.4. Theoretical Calculations

Geometry optimizations and frequencies calculations were carried out at the B3LYP-D3/6-31+G(d,p) with implicit solvent chlorobenzene (SDM) using Gaussian 09 [52]. Stationary points and transition states were confirmed by checking the absence of imaginary frequencies. The BDE(OH) values were determined by the isodesmic approach, from the total energy in solution computed by single point calculations, and by applying thermal correction to enthalpy.

### 3.5. Statistical Analysis

Each value was taken from at least three independent measurements, and results are expressed as an average. Errors for *n* and *k*_inh_ represent ± SD (SD = standard deviation).

## 4. Conclusions

In this work, the rate constants of the reaction of peroxyl radicals ROO• with honokiol **1**, its two derivatives **2** and **3**, and the other four bisphenol neolignans **4**–**7** were determined in apolar and polar solvents. The different hydroxyl and alkyl substitutions on the phenolic skeleton of the synthesized compounds affects their antioxidant activity, compared to that of the natural derivative honokiol **1**. The presence of alkyl groups in the *ortho*-position with respect to the -OH groups decreases the *k*_inh_ as well as the presence of 2-hydroxyethyl substituents. 4,4′-dihydroxybiphenylic structures increase the overall inhibition constant compared with honokiol, but lead to a decrease in the number of trapped radicals (*n* = 2 vs. *n* = 4). Compounds showing quinone-like oxidized forms (e.g., **1ox**, **5ox** and **7ox**) can be regenerated by exploiting the reducing effect of hydroperoxyl radicals generated by the addition of 1,4-cyclohexadiene to the reaction environment. This synergy occurs due to a catalytic cycle in which CHD acts as the sacrificial reductant, releasing HOO• radicals during the autoxidation and the consequent chain–transfer processes. For such quinones, the obtained antioxidant effect is enhanced if combined with HOO• radicals, rather than that of the starting compounds. The superior radical trapping activity of catechol derivative **7** and its ability to be regenerated by HOO• renders it an interesting molecule for further bioactivity studies.

Hopefully, the data presented herein will aid future investigation in the area, including the rational design of novel bioactive structures and possibly pharmacologically active lignans.

## Data Availability

The data presented in this study are available on request from the corresponding author.

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
