# Peer review of "Reaction with ROO• and HOO• Radicals of Honokiol-Related Neolignan Antioxidants"

_molecules, 2023, doi:10.3390/molecules28020735_

Round 1

Reviewer 1 Report

See in attached file

Reviewer 2 Report

This work extensively studies the antioxidant mechanism of bisphenols in the autoxidation processes initiated by peroxyl and hydroperoxíl radicals. The results are based of the slower rate of oxygen consumption in the presence of inhibitors. The inhibition efficiency was found two orders of magnitude larger for compound 7 compared to the other bisphenols. This fact was related toy the lower BDE values, but the authors should also analyze the impact of the number of OH groups in the phenyl rings, since this is 3 for compound 7, while only 2 for the other compounds.

The paper offers useful information how the presence of alkyl group can modify the inhibition rates, and suggest a regeneration process of phenols from the oxidized form by generating hydroperoxyl radicals.

The rate constants were evaluated by Eq. (8), but the derivation of this equation is not analyzed, only a reference is given to the experimental section. A detailed explanation should be gived for Eq. 8.

In Figure 3 the kinetics for compound 7 is given at half cc compared to the other inhibitors. The same cc should give better comparison.

In Table 3 the numbering of OH groups is replaced. This should be discussed.

Author Response

Response to Reviewer 2 Comments

This work extensively studies the antioxidant mechanism of bisphenols in the autoxidation processes initiated by peroxyl and hydroperoxíl radicals. The results are based of the slower rate of oxygen consumption in the presence of inhibitors. The inhibition efficiency was found two orders of magnitude larger for compound 7 compared to the other bisphenols. This fact was related toy the lower BDE values, but the authors should also analyze the impact of the number of OH groups in the phenyl rings, since this is 3 for compound 7, while only 2 for the other compounds.

The paper offers useful information how the presence of alkyl group can modify the inhibition rates, and suggest a regeneration process of phenols from the oxidized form by generating hydroperoxyl radicals.

Point 1: The rate constants were evaluated by Eq. (8), but the derivation of this equation is not analyzed, only a reference is given to the experimental section. A detailed explanation should be gived for Eq. 8.

Response 1:

Equation 8 was first developed by Denisov and derives from the classical kinetic equations describing lipid peroxidation, but it does not neglect the peroxyl radical self decay (2 ROO• à products). We described the use of this equation in detail herein: doi.org/10.1155/2017/6369358. This equation is described also here: DOI: 10.1039/b105079f, DOI: 10.1039/a904899e, E. T. Denisov and I. V. Khudyakov, Chem. Rev., 1987, 87, 1313; 10.1021/cr00082a003, E. T. Denisov, Liquid Phase Reaction Rate Constants, Plenum, New York, 1974. 10.1007/978-1-4684-8300-0

For more explanation, we have therefore decided to add these additional references [32-36] in line 191.

Point 2: In Figure 3 the kinetics for compound 7 is given at half cc compared to the other inhibitors. The same cc should give better comparison.

Response 2:

For compound 7 we decided to carry out these experiments at this concentration because a slope of the inhibited tract is crucial to measure the inhibition constant. At a higher concentration, no oxygen consumption would be observed, and it would be impossible to measure the inhibition constant. Moreover, with this concentration the picture is more clear and there is less overlap with the other lines (the red line would be very similar to the blue line).

Point 3: In Table 3 the numbering of OH groups is replaced. This should be discussed.

Response 3:

The numbers of the OH in table 3 are only used to refer the BDE to a specific OH group. We also corrected an error of numbering in compound 4.

Reviewer 3 Report

Research article provides new information about antioxidant properties of honokiol inspired neolignans, that were synthesized. Methods are described in detail. Also in results and discussion part, the authors taking into account limitation of their work, for example solubility of the samples, addition of methanol adn duo to that fact reevaluation of results. The schemes and figures in the article are well described, they are detailed and clear. Important is to continue in this study with further bioactive studies of newly synthesized compounds. The used literature is sufficient, it also contains articles from the last 5 years.

Suggestions:

1.      I suggest to reorganize article, to move section Results and Discussion after the section Material methods. Authors described their results and discussion directly after introduction part. More clear is to move it after material and methods section of the manuscript.

Typical organization of research article is also mentioned in Authors instructions: „The structure should include an Abstract, Keywords, Introduction, Materials and Methods, Results, Discussion, and Conclusions (optional) sections“

2.      In the introduction section, I would suggest to add scheme of shikimic acid pathway. Also authors mentioned the classification of lignans to 8 groups and neolignans to 15 groups. Addition of table about mentioned classification would enrich the article. 

3.      Why did the authors choose pinoresinol, arctigenin from the class of lignans and specify their structure and properties? How does it relate to the issue/topic? 

4.      Authors listed, that honokiol and magnolol „These two bisphenolic neolignans are present in the bark of Magnolia officinalis and other M. spp, whose extracts have been employed in oriental med icine to treat several disorders [11] Which specific diseases do they mean? 

5.      Which specific spectroscopy methods were used to characterize newly synthesized compounds? 

6.      Table 3: is about oxygen consumption during the autoxidation of Cumene (3.6 M) initiated by AIBN (0.05 M) in PhCl at 30° C.

It is stated that authors used half the concentration of compound 7 (0.7x10-5M) compared to the others (1.4x10-5M). What is the reason?

7.  In Table 1 is stated that the values are average, and in the section Statistical Analysis authors explain, that results are expressed as mean. Mathematically, average and mean are similar to each other as it is used to explain the set of numbers. I recommend to state it uniformly.

8.  I do not feel qualified about english, but I suggest to double check english, due to some typo mistakes as for example: 4- hydroxy-phneyl boronic acid, Subsequntly..

Author Response

Response to Reviewer 3 Comments

Research article provides new information about antioxidant properties of honokiol inspired neolignans, that were synthesized. Methods are described in detail. Also in results and discussion part, the authors taking into account limitation of their work, for example solubility of the samples, addition of methanol adn duo to that fact reevaluation of results. The schemes and figures in the article are well described, they are detailed and clear. Important is to continue in this study with further bioactive studies of newly synthesized compounds. The used literature is sufficient, it also contains articles from the last 5 years.

 Suggestions:

Point 1: I suggest to reorganize article, to move section Results and Discussion after the section Material methods. Authors described their results and discussion directly after introduction part. More clear is to move it after material and methods section of the manuscript.

Typical organization of research article is also mentioned in Authors instructions: „The structure should include an Abstract, Keywords, Introduction, Materials and Methods, Results, Discussion, and Conclusions (optional) sections“

Response 1:

We thank for the suggestion, but we prefer to leave the order of the sections in the present form.

We followed the instructions at https://www.mdpi.com/journal/molecules/instructions and we also used the recommended template

Point 2: In the introduction section, I would suggest to add scheme of shikimic acid pathway. Also authors mentioned the classification of lignans to 8 groups and neolignans to 15 groups. Addition of table about mentioned classification would enrich the article.

Response 2:

We thank the referee for this suggestion. We have not previously reported the shikimic acid pathway to avoid a too-long introduction. We have briefly summarised the biosynthetic pathway for the synthesis of dimers starting from phenyl propanoids C6C3 in the Supplementary Materials. In the manuscript, we have inserted two more figures with descriptions of the structures of lignans and neolignans accordingly to the classification proposed by Teponno et al.

Point 3: Why did the authors choose pinoresinol, arctigenin from the class of lignans and specify their structure and properties? How does it relate to the issue/topic?

Response 3:

In discussing the antioxidant properties of lignans and neolignans, which are characterized by a large variety of structure, we have decided to insert a few representative examples, which includes pinoresinol, a furofuran lignan, arctigenin a member of dibenzylbutyrolactone lignan, and magnolol and honokiol two bisphenol neolignans.

Point 4: Authors listed, that honokiol and magnolol „These two bisphenolic neolignans are present in the bark of Magnolia officinalis and other M. spp, whose extracts have been employed in oriental med icine to treat several disorders [11]“ Which specific diseases do they mean?

Response 4:

We thank the referee for this suggestion. We have now completed the sentence with the required information.

Point 5: Which specific spectroscopy methods were used to characterize newly synthesized compounds?

Response 5:

The newly synthesized compound 7 was characterized by 1D and 2D NMR spectroscopy and high-resolution mass spectrometry. The data are listed in the Materials and Methods section. However, we gave additional information in the general section. 

Point 6: Table 3: is about oxygen consumption during the autoxidation of Cumene (3.6 M) initiated by AIBN (0.05 M) in PhCl at 30° C.

It is stated that authors used half the concentration of compound 7 (0.7x10-5M) compared to the others (1.4x10-5M). What is the reason?

Response 6:

As explained above for Reviewer 2, due to the high kinh of compound 7 it is necessary to use a more dilute solution to perform the experiments, in order to observe a slope in the inhibited period. At a higher concentration, no oxygen consumption would be observed, and it would be impossible to measure the inhibition constant.

Point 7: In Table 1 is stated that the values are average, and in the section Statistical Analysis authors explain, that results are expressed as mean. Mathematically, average and mean are similar to each other as it is used to explain the set of numbers. I recommend to state it uniformly.

Response 7:

We are sorry for the mistake. We corrected the Statistical Analysis section.

Point 8: I do not feel qualified about english, but I suggest to double check english, due to some typo mistakes as for example: 4- hydroxy-phneyl boronic acid, Subsequntly..

Response 8:

Sorry for the mistakes. We have corrected them in the text.

Reviewer 4 Report

In the nanuscript   molecules-2101524 the authors describe their findings on the kinetics and  mechanisms of antioxidant activity of series of poliphenols belonging to the group of neolignans. This is a continuation of previous publications published in 2017 in Org Biomol Chem (with other isomeric lignin-derived compounds). The manuscript presents high quality and, in general, is well presented. The authors have experience in the techniques they are using,    and the hypotheses, verifications and interpretation of the results are correctly elaborated and well supported by the literature (if needed).

 There are some problems that need to be explained  and corrected. in the revised version of ms.

1. page 6, line 161: “Despite present in very small amount, methanol strongly affects the inhibition constant of Honokiol 1 in chlorobenzene by about 3 times (see Table 1).” This phrase is not clear and am not able to follow it – do the authors meas the rate constant presented in Table 1 (in chlorobenzene) versus rate constant from literature? Please clarify.

2. Scheme 3 and accompanying text (lines 172-176): I agree that in  chlorobenzene the reaction with second peroxyl radical  would be formal H atom transfer (as suggested by authors) however in acetonitrile (which is a solvent frequently used in electrochemistry as  an  ion solvating agent) the phenoxyl radical will be very strong acid. Phenoxyl is very strongly electron withdrawing group, see J. Am. Chem. Soc. 2022, 144, 47, 21783–21790, see also footnote 46 in J. Org. Chem. 2004, 69, 5888-5896, and a consequence should a fast deprotonation (even in acetonitrile) of the second, remaining hydroxyl group attached  to another, conjugated aromatic ring. Therefore, some hydroxyl groups will be deprotonated and those non-deprotonated will be H-bonded. For this reason, I do not exclude that the final product in acetonitrile also can be structure 1ox, as radical anion (phenoxyl radical in ring A and phenolate anion in ring B) will be a strong reducing agent for alkylperoxyl radicals (final product will be hydroperoxide and 1ox).

3. page 6 line 188:  not clear phrase (“In ACN, for the second phenolic ring, the stoichiometric coefficient could not be measured because the kinh value is too low. For this reason, we only found n = 2.”) Stoichiometric coefficient is rather overall value, do the aurthors distinguil=shed n for first ring and another n for second ring of the molecule? Are the values of n presented in Table 1 for 6 and 7 in acetonitrile for overall stoichiometric coefficient or are they estimated or taken from literature?

  Minor problems or suggestions

4. please use correct names ( pagebn2,line 79: peroxyl and hydroperoxyl, not peroxy and hydroperoxy)

5. equation 8:  please replace capital K into k in the  rate constants inhibition (kinh) and termination   kt.

caption to Scheme 4: n is 4, whereas in Table 1 n=3.8. Please unify.

6. Structures in Table 3: please connect the dashed bonds (in structures 1-3) to the center of aromatic ring). By the way, what is  the angle between the planes of aromatic rings?  And how to explain that  hydroxyl group  marhes as 1 is weaker than hydroxyl group marked as 2: group 1 is internally H-bonded to aromatic ring, so, it should be regarded as second in the order of H atom abstraction.

7. references: please CAREFULLY check all bibliographic  details because I have found  some errors in ref 31  (year is not correct), and in ref 36 (not correct name of the journal). If Endnote was used, please check endnote formatting for all references (or add DOI, for easy identification of the source)   

Author Response

Response to Reviewer 4 Comments

In the manuscript  molecules-2101524 the authors describe their findings on the kinetics and  mechanisms of antioxidant activity of series of poliphenols belonging to the group of neolignans. This is a continuation of previous publications published in 2017 in Org Biomol Chem (with other isomeric lignin-derived compounds). The manuscript presents high quality and, in general, is well presented. The authors have experience in the techniques they are using,    and the hypotheses, verifications and interpretation of the results are correctly elaborated and well supported by the literature (if needed).

There are some problems that need to be explained  and corrected. in the revised version of ms.

Point 1: page 6, line 161: “Despite present in very small amount, methanol strongly affects the inhibition constant of Honokiol 1 in chlorobenzene by about 3 times (see Table 1).” This phrase is not clear and am not able to follow it – do the authors meas the rate constant presented in Table 1 (in chlorobenzene) versus rate constant from literature? Please clarify.

Response 1:

We clarified better this point on line 209.

Point 2: Scheme 3 and accompanying text (lines 172-176): I agree that in  chlorobenzene the reaction with second peroxyl radical  would be formal H atom transfer (as suggested by authors) however in acetonitrile (which is a solvent frequently used in electrochemistry as  an  ion solvating agent) the phenoxyl radical will be very strong acid. Phenoxyl is very strongly electron withdrawing group, see J. Am. Chem. Soc. 2022, 144, 47, 21783–21790, see also footnote 46 in J. Org. Chem. 2004, 69, 5888-5896, and a consequence should a fast deprotonation (even in acetonitrile) of the second, remaining hydroxyl group attached  to another, conjugated aromatic ring. Therefore, some hydroxyl groups will be deprotonated and those non-deprotonated will be H-bonded. For this reason, I do not exclude that the final product in acetonitrile also can be structure 1ox, as radical anion (phenoxyl radical in ring A and phenolate anion in ring B) will be a strong reducing agent for alkylperoxyl radicals (final product will be hydroperoxide and 1ox).

Response 2:

We thank the Reviewer for this observation. However, the acidity of semiquinones is reported to be similar to acetic acid in water, therefore they behave as (relatively) weak acids as they have pKa in the range 4-5. In MeCN, the pKa are much higher than in water, and thus semiquinones are expected to be not ionized.

Point 3: page 6 line 188:  not clear phrase (“In ACN, for the second phenolic ring, the stoichiometric coefficient could not be measured because the kinh value is too low. For this reason, we only found n = 2.”) Stoichiometric coefficient is rather overall value, do the aurthors distinguil=shed n for first ring and another n for second ring of the molecule? Are the values of n presented in Table 1 for 6 and 7 in acetonitrile for overall stoichiometric coefficient or are they estimated or taken from literature?

Response 3:

When possible, we distinguished between the stoichiometry of the two OH groups and this is mentioned in Table 1. The values for compounds 5, 6 and 7 are calculated from ours experiments and are not known in literature.

  Minor problems or suggestions

Point 4: please use correct names ( pagebn2,line 79: peroxyl and hydroperoxyl, not peroxy and hydroperoxy)

Response 4:

We thank the referee. Done

Point 5: equation 8:  please replace capital K into k in the rate constants inhibition (kinh) and termination   kt.

caption to Scheme 4: n is 4, whereas in Table 1 n=3.8. Please unify.

Response 5:

We thank the referee. We have fixed the mistakes

Point 6: Structures in Table 3: please connect the dashed bonds (in structures 1-3) to the center of aromatic ring). By the way, what is  the angle between the planes of aromatic rings?  And how to explain that  hydroxyl group  marhes as 1 is weaker than hydroxyl group marked as 2: group 1 is internally H-bonded to aromatic ring, so, it should be regarded as second in the order of H atom abstraction.

Response 6:

As correctly suggested, we connect the dashed bond to the center of the aromatic ring.

In a previous paper, we reported the angle between the aromatic rings as about 50° in the case of honokiol (however, the angle changes between the phenol and the radical form) and about 30° for para-bisphenols.

The H-bond to the aromatic ring indeed is very weak. IR experiments already reported by us for honokiol show that it is weaker than the H-bond with the allylic double bond.

Point 7: references: please CAREFULLY check all bibliographic  details because I have found  some errors in ref 31  (year is not correct), and in ref 36 (not correct name of the journal). If Endnote was used, please check endnote formatting for all references (or add DOI, for easy identification of the source)  

Response 7:

We thank the referee. We have fixed the mistakes